# Whole Genome Sequencing of A(H3N2) Influenza Viruses Reveals Variants Associated with Severity during the 2016–2017 Season

**DOI:** 10.3390/v11020108

**Published:** 2019-01-28

**Authors:** Bruno Simon, Maxime Pichon, Martine Valette, Gwendolyne Burfin, Mathilde Richard, Bruno Lina, Laurence Josset

**Affiliations:** 1Virpath, CIRI, Univ Lyon, Inserm U1111 CNRS UMR5308, ENS, UCBL, Faculté de Médecine Lyon Est, 69372 Lyon, France; sib0.smb@gmail.com (B.S.); Maxime.PICHON@chu-poitiers.fr (M.P.); martine.valette@chu-lyon.fr (M.V.); bruno.lina@chu-lyon.fr (B.L.); 2Laboratoire de Virologie, IAI, CBN, Groupement Hospitalier Nord, Hospices Civils de Lyon, 69317 Lyon, France; 3Centre National des Virus Respiratoires, IAI, CBN, Groupement Hospitalier Nord, Hospices Civils de Lyon, 69317 Lyon, France; gwendolyne.burfin@chu-lyon.fr; 4Department of Viroscience, Erasmus MC, 3000 Rotterdam, The Netherlands; m.richard@erasmusmc.nl

**Keywords:** influenza, NGS, severity, vaccination, epidemiology, quasispecies

## Abstract

Influenza viruses cause a remarkable disease burden and significant morbidity and mortality worldwide, and these impacts vary between seasons. To understand the mechanisms associated with these differences, a comprehensive approach is needed to characterize the impact of influenza genomic traits on the burden of disease. During 2016–2017, a year with severe A(H3N2), we sequenced 176 A(H3N2) influenza genomes using next generation sequencing (NGS) for routine surveillance of circulating influenza viruses collected via the French national influenza community-based surveillance network or from patients hospitalized in the intensive care units of the University Hospitals of Lyon, France. Taking into account confounding factors, sequencing and clinical data were used to identify genomic variants and quasispecies associated with influenza severity or vaccine failure. Several amino acid substitutions significantly associated with clinical traits were found, including NA V263I and NS1 K196E which were associated with severity and co-occurred only in viruses from the 3c.2a1 clade. Additionally, we observed that intra-host diversity as a whole and on a specific set of gene segments increased with severity. These results support the use of whole genome sequencing as a tool for the identification of genetic traits associated with severe influenza in the context of influenza surveillance.

## 1. Introduction

Seasonal influenza infections remain a considerable health and socioeconomic challenge, as vaccine uptake and effectiveness are still limited. The resulting morbidity and mortality have only marginally decreased in recent years, with annual epidemics still causing around 290,000 to 650,000 respiratory deaths worldwide [1]. This continuing burden leads to the constant adaptation of public health measures, including vaccination policies which remain the most potentially effective method to limit the impact of influenza [2]. However, current vaccines have moderate effectiveness, in part because of major and minor mismatches between vaccine and circulating influenza viruses, and weak immune responses elicited in the elderly. Severe influenza occurs most frequently in certain demographic groups, including the very young and the very old, but also those suffering from chronic respiratory, cardiac, and metabolic diseases, as well as pregnant women and obese patients, as shown for the A(H1N1) pdm09 virus [3,4]. Antivirals that control the infection in some patients at risk of complications are available and usually recommended, yet resistant strains often appear in cases of prolonged treatments. In this context, influenza surveillance is mostly focused on studying antigenic variants to select vaccine strains [5] or resistance mutations to monitor antiviral effectiveness [6,7]. Many mutations in the influenza genome have been linked to virulence but clinical studies are scarce and have had, apparently, a minimal impact on clinical management, unlike what is routinely done in the case of bacterial infections [8]. This may be in relation to influenza pathogenesis that is dependent on complex and still-unpredictable mechanisms, including multigenic equilibrium and interactions within the viral population and with the host immunity, in the context of a fast-evolving virus with a highly error-prone replication. Moreover, influenza substitutions can only be considered within a specific viral background, as it was shown that their impact differed even among strains of the same subtype [6]. Therefore, global approaches considering virological and immunological parameters associated with clinical severity, as a whole, are better suited to investigate influenza pathogenesis [9,10].

Recently, the improved accessibility of next generation sequencing technologies has allowed their use in contexts closer to routine clinical practice such as in University hospital centers and national surveillance networks. This has led to considering each infection as being caused by a swarm of nearly identical viruses (or quasispecies) rather than by an individual strain. For influenza, quasispecies were found to have an important role in the pathophysiology of infections and to be involved in virulence and intra-host and inter-host spread [11,12,13,14,15]. Several beneficial characteristics of quasispecies at the host scale have been reported [11]. For example, it was shown that minor resistance variants pre-exist any treatment despite lower fitness and that, at least in vitro, a cooperation between variants for an improved global fitness exists [16,17]. A(H3N2) viruses are considered to evolve faster than other subtypes [18,19]; in recent years they have been found to have high clade diversity and increased morbidity and mortality [20,21]. During the 2016–2017 influenza epidemic, the majority of influenza viruses circulating in the northern hemisphere were A(H3N2) viruses from clade 3c.2a. In France, the dominance of A(H3N2) viruses was associated with an early season of moderate intensity, but with a high severity particularly among the elderly in a context of low vaccine coverage and sub-optimal vaccine effectiveness [22]. Low vaccine effectiveness was related to a hemagglutinin mutation in the 2016–2017 egg-adapted H3N2 (clade 3c.2a) vaccine strain A/HongKong/4801/2014 that altered its antigenicity [23]. As a consequence, there is a growing interest in performing in-depth genomic analysis of these H3N2 viruses, including at the quasispecies level.

We describe herein the genomic characteristics of Influenza A (H3N2) clinical strains recovered via the French national influenza surveillance network or from patients hospitalized in the intensive care units of the University Hospitals of Lyon, France. Several substitutions were significantly associated with severe cases and/or ineffective vaccination, some of them only in specific backgrounds. Individually, minor variants did not appear to be associated with any clinical traits, whereas the intra-host diversity as a whole and on a specific limited set of gene segments increased with severity.

## 2. Materials and Methods

### 2.1. Ethics Statement

This is a retrospective study, requiring no additional samples from patients as they were obtained during routine diagnosis. This study is considered non-interventional, as clinical and therapeutic data collection was performed retroactively from a clinical files database, respecting patient confidentiality. This study was approved by the ethics committee of the Lyon University Hospitals (Hospices Civils de Lyon, France), on 3 May, 2017.

### 2.2. Clinical Samples

Samples included in this study were recovered between May 2016 and April 2017 via the French national influenza surveillance network or from patients hospitalized in Lyon University Hospitals. This cohort was designed to investigate molecular evolution of influenza viruses and studied herein in a secondary analysis. The French national influenza surveillance network recruits patients with influenza-like illnesses in ambulatory care and in University hospitals. Only data from the southern half of France were included. In the Lyon University hospitals systematic surveillance was set up for influenza cases hospitalized in the intensive care units and for influenza in vaccinated patients. When several samples were retrieved and/or sequenced for a patient, only the earliest sampled after the onset of symptoms was included. For the present study, only treatment-naïve patients infected by the A(H3N2) strains for which we obtained at least partial sequences for each genomic segment were included. Clinical data were retrieved from questionnaires filled out by the healthcare professionals who took the samples and from computerized clinical files for the surveillance network and Lyon University Hospitals, respectively. All patients had sufficient clinical data reported to classify them as having either mild or severe influenza. Clinical criteria leading to classification of patients as severe included admission to intensive care for respiratory complications, need for continuous supplemental oxygen or hypoxemia, neurological signs (seizures, fainting with loss of consciousness, evocative imagery, or lumbar puncture) or any type of shock. Severe cases without any major respiratory signs were subclassified depending on their main symptomatology (Table 1).

### 2.3. RNA Extraction, Viral Load Determination, and Full-Genome Amplification

The following protocol was followed for all included samples, irrespective of source, sequencing date, or patient origin (Lyon University Hospital or French national influenza surveillance network). RNA was extracted from clinical respiratory samples conserved at −20 °C using Nuclisens EasyMag automated extraction platform (bioMérieux, Marcy-l’Etoile, France). In order to lower human and bacterial DNA content of the extract, nucleic acids were then incubated with 1 μL of Turbo DNAse, 2 μL of Turbo DNAse buffer (Life Technologies, Carlsbad, CA, USA), and 0.5 μL of RNasin plus RNase Inhibitor (Promega Corporation, Madison, WI, USA) for 90 min at 37 °C before stopping the enzymatic reaction through immediate purification by magnetic beads (0.5×; NucleoMag^®^ NGS Clean-up and Size Select, Macherey–Nagel, Düren, Germany). Each segment was retro-transcribed and amplified using a published multi-segment reverse transcription protocol, optimized by adding 0.5 μl of RNasin plus RNase Inhibitor [24].

### 2.4. Illumina Sequencing

The DNA library was prepared using the commercial Nextera™ XT DNA library kit (Illumina, San Diego, CA, USA) consisting of enzymatic tagmentation, barcoding by PCR (12 cycles), and size-selection using magnetic beads. Then, the sequencing run was performed on a NextSeq500 using a mid-output cartridge resulting in 2 × 150 nucleotides paired-end reads. Fluorescent images were analyzed using the Illumina RTA1.8/SCS2.8 base-calling pipeline to obtain demultiplexed FASTQ-formatted sequence data.

### 2.5. Bioinformatic Analysis

As described previously, the sequencing files were cleaned from human reads using bmtagger (v1.1.0) with human genome database hg19 [25,26]. Adapters and low quality ends of reads (Phred score <20) were removed by cutadapt (v0.4.4), also removing reads smaller than 50 bases and the paired reads associated with them [27]. The BWA-MEM alignment tool (v0.7.15) was used to verify sequenced strain subtypes by hemagglutinin (HA) homology, and then to map the reads on the circulating A(H3N2) strains’ common ancestor A/Perth/16/2009 [28]. This step generated consensus sequences that were used as the patients’ own mapping reference in further steps. All read mapping by BWA-MEM was performed using default settings, except for gap opening and extension penalties which were set to 10 and 2, respectively, to increase indel stringency. Sorted mapped reads were manually inspected using FastQC (v0.11.5) to verify read quality. Variant calling files were generated with naive variant caller (Biomina Galaxy platform) reporting all necessary base metrics for each position on the A(H3N2) genome, ignoring low-quality bases [29]. A homemade vcf analysis python script applied filters to validate possible single nucleotide polymorphisms and generate end-user workable files. These filters comprised a quantitative strand bias (SB) score estimating an uncertainty for variant frequencies when compared to the consensus bases disequilibrium and a depth-adapted threshold. This depth-adapted threshold was calculated following a t-test distribution derived from the commonly-approved 100X threshold necessary to exclude heterozygote mutations and was verified empirically [25]. We determined minimal depth thresholds for variant calling as 2000 and 10,000 reads for variants above 5% and 1% frequencies, respectively, and a relative strand disequilibrium lower than the variant estimated frequency.

### 2.6. Statistical Analysis

Statistical differences between experimental groups were analyzed using chi-square tests for amino acid substitutions. The diversity analyses were performed using the Wilcoxon rank sum test comparing the count of minor variants on the nucleotide level for each segment. In each case, *p*-values < 0.05 were considered significant. Phylogenetic trees were generated using the Mega7 Neighbor-Joining method, as performed by the European center for disease prevention and control, and using the Poisson correction method and by considering deletions pairwise [30]. Reassortment events were identified using the Graph-incompatibility-based Reassortment Finder (GiRaF) (v1.02), considering an event as validated if three pairwise segment comparisons supported the hypothesis with a confidence >0.95 [31].

### 2.7. Accession Numbers

The Bioproject accession number for the raw sequences is SRPXXXXXPRJNA516781 and all genomic sequences used in this study are available on the GISAID databank.

## 3. Results

### 3.1. Sequencing Efficacy

Among the 176 samples, 155 whole A(H3N2) consensus genomes were successfully sequenced when considering only those with more than 95% coverage. Overall, sequencing efficacy varied as a function of segment size; shorter segments were sequenced to a greater depth. However, sequencing depth was constantly low on two thirds of segment 2 (coding for the polymerase basic protein 1, PB1) and, for some patients, lower on the central part of viral polymerase coding segments (segments 1–3; Figure 1). After excluding segment 2, a mean depth on all segments above 10,000X was obtained for 78 A(H3N2) genomes, and a mean depth of 2000X for 142 A(H3N2) genomes. These depths of coverage allow calling variants above 1% and 5% frequencies, respectively.

### 3.2. Distribution of Consensus Variants According to Clades, Severity, or Vaccination

Substitution associations with severity or vaccine failure were sought. All 155 A(H3N2) whole genome reads sufficiently covered for the generation of consensus sequences were used in phylogenetic and consensus variant analysis. Among them, 121 viruses belonged to genetic clade 3c.2a1, eight belonged to 3c.2a2, 24 belonged 3c.2a3, one belonged to 3c.2a4, and one belonged to genetic clade 3c.3a (the only virus not part of the 3c.2a clade; Figure 2). Within clade 3c.2a1, different subgroups could be further defined based on additional HA substitutions. One virus belonged to the recently-defined subclade 3c.2a1a (T135K, HA2 G150E); 32 viruses belonged to a clade 3c.2a1 subgroup, close to vaccine strain A/Singapore/Infimh-16-0019/2016, defined by N121K and, for 14 of them, by G150E (HA2 numbering); 88 viruses belonged to another clade 3c.2a1 subgroup, close to A/England/919/2016 (a Bolzano-like strain), whether defined by R142G or other various substitutions (A/England/919/2016 cluster; Figure 2).

Overall, severe, vaccinated, and Lyon University Hospitals cases were broadly distributed, period- and clade-wise. None of the well-described treatment resistance substitutions (neuraminidase (NA): E119V, D151E, I222V, R224K, E276D, R292K, and R371K) [32] were present in any consensus genome, except for the conserved amantadine resistance substitution S31N. However, several substitutions across the whole genome could significantly distinguish mild from severe cases and vaccinated from non-vaccinated cases. These substitutions were distributed on the whole influenza proteome, however most were on HA (*n* = 8), NA (*n* = 7), and non-structural protein 1 (NS1; *n* = 4). Substitutions associated with vaccination status exclusively (*n* = 19) were more varied but had weaker statistical significance than those exclusively associated with severity. For the latter, two substitutions, V263I on NA and K196E on NS1, were particularly significant (*p* < 0.0001; Table 2a). As represented in Figure 2, NA V263I and NS1 K196E were mostly found in a specific 3c.2a1 clade background (as defined by HA phylogeny). The NA substitution was only present alone in two strains associated with mild disease outside this background, whereas the NS1 substitution was not necessarily in NA-substituted strains. However, when co-occurrent, these mutations distinguished mild from severe cases more precisely (Table 2b). Based on 3D structures, it could be noted that some of the following substitutions were found: On HA positions 131 and 144 which are close to its receptor binding site (Figure 3a), on NS1 p85β binding site positions 99 and 146 (Figure 3b), and on NA position 149 which is close to its sialidase active site (Figure 3c). Additionally, reported substitutions were verified as not being linked to depth-specific features of the respiratory tract sites. When comparing Lyon University Hospital cases, comprising both lower and higher respiratory tract samples (50 and 21 samples included in the analysis, respectively), none of the reported substitutions were associated with sample type.

HA clustering of the strains did not seem to originate from within-subtype reassortments, possibly implying the substitutions could have hitchhiked alongside another event rather than being implicated directly in observed correlations. Phylogenetic trees generated from proteins other than HA found that the same strains clustered consistently, however without being similar to such an extent as to indicating a possible common index case. When verified by GiRaF, only one strain was detected as possibly resulting from a reassortment event. This strain, A/Poitiers/917/2017, presented genomic segments closely related to A/Switzerland/9715293/2013, the 3c.3a reference strain, except for its NS1 coding segment corresponding to a 3c.2a1 genotype.

### 3.3. Diversity and Quasispecies Analysis

When analyzed individually, no significant role of minor variants was observed in distinguishing the clinical groups. These minor variants were rarely at the same positions as the consensus substitutions. An exception to this was found on the HA. The HA 160K substitution, resulting in the loss of a glycosylation site for these quasispecies, was found in 14 cases (with frequencies ranging from 1.2% to 98%, median 7.1%) and was significantly associated with severity (*n* = 11; *p* < 0.05). Otherwise, the same minor variant was rarely found in several patients and at most in three patients (*n* < 4) and a major part of polymerase-coding segment diversity corresponded to deletions. In order to discern potential associations between viral particle diversity and clinical features, we had to perform the analysis of global diversity by segment in each case. Considering that sequencing depth was variable, groups were defined according to segment and targeted depth threshold. The mean diversities of segments 3, 4, and 6 (coding for polymerase acidic (PA), HA, and NA proteins, respectively) were significantly higher for severe cases than in mild cases (Figure 4).

Note that a considerable number of mutants had frequencies between 1% and 5%, thus lowering the depth threshold to 1% was necessary to notice the diversity difference for segment 6. In comparison, diversity did not vary significantly with vaccination status (Table 3). Once more, as a control, diversity was verified as not being significantly associated with respiratory tract sampling depth in the cases from the Lyon University Hospitals. 

## 4. Discussion

In the present study we conducted a comprehensive genomic characterization of H3N2 viruses from either the French national influenza surveillance network or the Lyon University Hospitals’ intensive care wards using a combination of several approaches. This allowed the impact of antigenic proximity, genomic substitutions, quasispecies diversity, and reassortment events to be studied concomitantly, and the results therefore provide a comprehensive view of mechanisms involved in the severity of the 2016–2017 seasonal influenza in France.

These data were generated from a secondary analysis of a cohort designed to investigate molecular evolution of the influenza virus. The size of the study did not permit the performance of multivariate analysis correcting for sample nature, comorbidities, geographic, or temporal confounding factors that could influence influenza severity, a limitation that is shared with previous studies [8]. Furthermore, most severe cases were, by design, from the same region albeit with several hospital centers. However, sequencing was performed directly on clinical samples, avoiding any cell culture or experimental infection bias [33,34]. Clinical samples from this study were collected exclusively from treatment-naïve patients. They were fairly equally distributed between mild and severe cases, and can be considered to be representative of the spectrum of influenza diseases in the 2016–2017 season. It should be noted that the specimens of surveillance network patients were mostly collected in the upper respiratory tract and earlier after the onset of symptoms than those from hospitalized cases. However, there was no significant difference according to site of respiratory tract sampling within the Lyon University Hospital cases, as this group comprised both upper and lower respiratory tract sample types which could be a source of diversity by itself or by being correlated to viral copy number. In addition, we did not find any substitutions in the HA receptor binding site contrary to what could be expected if influenza viruses’ mutations were driven by location on the respiratory tract. Nonetheless, these results should be interpreted as hypothesis generating and future studies validating the impact of mutations described here are warranted.

Several substitutions were associated with severity or vaccination status in the study population. Among these, most had not been extensively studied in vitro or in vivo, hence determining their actual involvement would need further investigations. The two most significant ones, NA V263I and NS1 K196E (*p* < 0.001), have previously been described in other contexts and were strongly associated with the severe group. Note that these substitutions were mostly found in strains belonging to a subgroup of the 3c.2a1 clade. This subgroup differed from the others in the 3c.2a1 clade by not harboring commonly found substitutions HA1 N121K and HA2 G150E. Interestingly, both substitutions were reported co-varying alongside substitutions altering p85β binding by NS1 in H1N1 strains [35]. Additionally, the substitution NS1 K196E was reported to increase virulence in mice when induced in an H5N1 backbone [36]. To the best of our knowledge, NA V263I has not been studied in an H3N2 background yet. However, a comprehensive study of NA V263I involvement at the structural level suggested the isoleucine, which is two methyl groups higher than a valine, could allow important changes by altering hindrance effects [37]. It was also suggested that this site was part of an epitope that was suitable for therapeutic monoclonal antibodies, and that an amino acid change at this position modified the immune response [38]. We also reported that a substitution on V149A on NA, which is close to the active site, may have some impact on the sialidase activity. Substitutions significantly associated with vaccination status comprised many clade-defining HA substitutions (HA1 N121K, S144K, N171K, HA2 I77V, G150E), reflecting strains antigenically less related to A/Hong Kong/4801/2014, the 2016–2017 H3N2 vaccine strain. Notably, substitutions were found on position 131 and 144 close to positions 155 and 145 which have been shown to be crucial for the evolution of H3N2 viruses [5]. As vaccination efficacy relies on the antigenic proximity with vaccinal strains, these associations with putative antigenic changes, albeit statistically weaker, make sense and tend to support the validity of the results presented herein.

Next generation sequencing (NGS) has brought to light new areas of research in microbiology, in particular on genetic diversity and on the implication of quasispecies in physiopathology [39,40,41,42,43,44]. For instance, certain influenza virus quasispecies have been reported fostering cell-to-cell mechanisms under treatment pressure [13], but also changing viral tropism [8]. When analyzed on a global genomic diversity scale, we found an association between severity and high diversity whereas vaccination status had no impact, as previously described [45,46]. Herein, we also found that this association concerned segments 3, 4, and 6 (coding for PA, HA, and NA respectively) only. This is likely to be in relation to specific selective pressure experienced by each segment that is mainly dependent on the “swarm” needs, viral particles constitution, and coded proteins [11]. For now, hypothesizing why these segments show a significant difference would not be supported by any experimental evidence. Thus, we believe genetic diversity should be considered for each segment independently. As explained in the sequencing efficacy description of the samples herein, an inconsistent read depth was observed on polymerase-coding segments, which could suggest the existence of defective particles in certain cases [15]. We could not conclude the reliability of these observations as the study was not designed to explore this complex problem. An approach focused on anticipating the difficulties related to molecular biology and bioinformatic analysis in this case is required.

In conclusion, the global approach of influenza sequencing results reported herein indicate that multiple factors need to be considered to understand the causes of the 2016–2017 season epidemic severity and support the potential added value of whole genome sequencing for influenza surveillance.

## Figures and Tables

**Figure 1 viruses-11-00108-f001:**
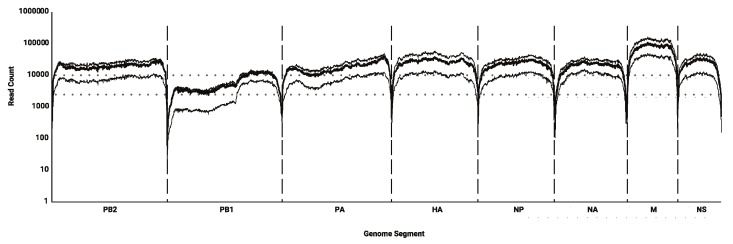
Segment coverage of A(H3N2) influenza virus genomes. Mean depth line for each segment is framed by thinner lines corresponding to the first and third quartiles of depth of coverage. Minimal depth thresholds for 5% (2000X) and 1% (10,000X) variant calling are represented by dot lines.

**Figure 2 viruses-11-00108-f002:**
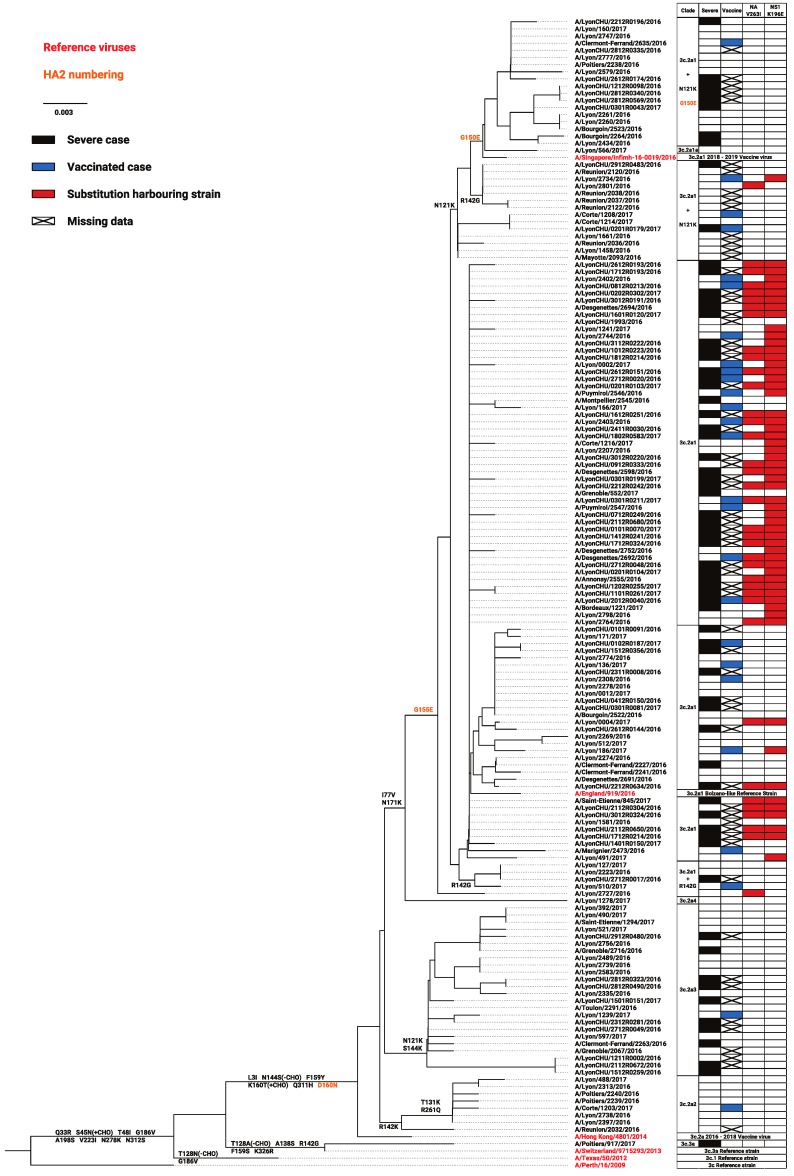
Phylogenetic tree constructed using hemagglutinin genes from the 155 subjects of the 2016–2017 season for whom the whole influenza consensus genome was successfully sequenced and representatives of the A(H3N2) clades. Every substitution concerning antigenic sites were reported on branches. Neuraminidase (NA) V263I or non-structural protein 1 (NS1) K196E presence, severity, and vaccination status of the patient they are associated with, and the clade in which they belong, is reported next to each strain.

**Figure 3 viruses-11-00108-f003:**
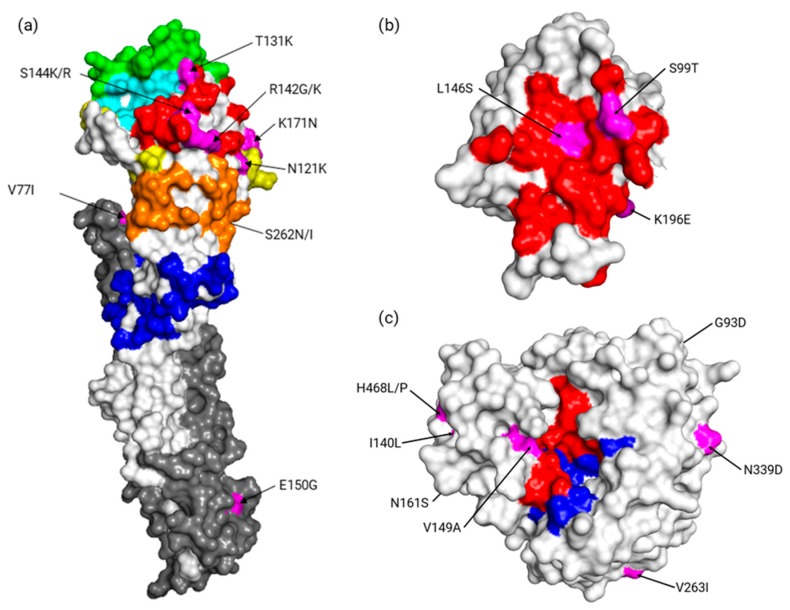
Contextualized locations of substitutions described in our study (in magenta). (**a**) On HA (PDB accession no. 4O5N) with antigenic sites A (red), B (green), C (blue), D (yellow), and E (orange) highlighted and the receptor binding site in cyan. (**b**) On NS1 (PDB accession no. 3EE8) with the p85β-binding site colored red. (**c**) On NA (PDB accession no. 6BR6) with sialidase catalytic residues in red and framework residues in blue. The image was generated with PyMOL software (Delano Scientific, San Carlos, CA, USA).

**Figure 4 viruses-11-00108-f004:**
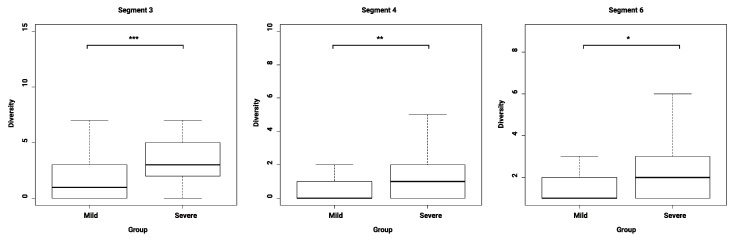
Box plots comparing viral diversity, when considering variants above 1%, for segments with a significant *p*-value. The whiskers of the boxplots show the range of the data comprised in the first and last quartiles. * *p* < 0.05, ** *p* < 0.005, *** *p* < 0.001.

**Table 1 viruses-11-00108-t001:** Included patients baseline demographics and clinical and sequenced sample characteristics.

	Patient Characteristics		Mild Outcome	Severe Outcome
**Baseline Demographics**	Number of patients (% of total)		97 (55%)	79 (45%)
	Age in median years (range)		34 (0–91)	73 (1–97)
	Sex	Male	48	36
		Female	48	43
	Origin	Lyon University Hospitals	6	65
		Surveillance network	91	14
**Sample Characteristics**	Sample type	NPA	2	6
		NS	95	49
		TBA	0	13
		BAL	0	11
	Median time since onset of symptoms, days (range)		1 (0–5)	3 (0–8)
	Viral load—median cycle threshold (range)		19.7 (14.9–30)	23.3 (17–35.3)
**Clinical Characteristics**	Vaccinated against Influenza for the current season		21	7
	Severe outcome risk factor *		17	65
	Main severity component	Respiratory	-	70
		Neurological		5
		Multiple organ failure		4

* Risk factors included age >65 years, immunodeficiency, ongoing cancer, chronical pulmonary, cardio-vascular, or neuro-muscular disease, BMI >35, diabetes, and ongoing pregnancy. NPA: Nasopharyngeal aspirate; NS: Nasal swab; TBA: Tracheobronchial aspirate; BAL: Bronchoalveolar Lavage.

**Table viruses-11-00108-t002a:** 

(a)																		
	**PB2**		**PB1**		**PA**		**HA1**							**HA2**			**NP**	
	**255**	**480**	**52**	**614**	**565**		**121**	**131**	**142**	**144**	**171**	**262**		**77**	**150**		**450**	**472**
Significant *p* <	0.05	0.05	0.05	0.05	0.05		0.05 0.05	0.05	0.01	0.05	0.05	0.05		0.01	0.01		0.05	0.05
Consensus	V	V	R	E	V		N	T	R	S	K	S		V	E		G	A
Variants	I	I	W/K	D	M		K	K	G/K	K/R	N	N/I		I	G		S/N	T
Severe cases	-	-	-	-	-		N	T	R	-	-	-		-	-		-	-
Vaccinated cases	V	V	W/K	D	M		N	-	-	S	K	S		V	E		G	A
	**NA**								**M1**		**M2**	**NS1**						**NS2**
	**93**	**140**	**149**	**161**	**263**	**339**	**468**		**147**		**23**	**99**	**146**	**196**		**224**		**67**
Significant *p* <	0.05	0.01 0.005	0.001 0.001	0.05	1 × 10^−4^	0.05	0.05		0.05		0.05	0.05	0.05	1 × 10^−4^ 0.001		0.05		0.05
Consensus	G	I	V	N	V	N	H		V		S	S	L	K		R		E
Variants	D	L	A	S	V	D	L/P		M		N	T	S	E		S/K		A/K
Severe cases	-	I	A	-	I	-	-		-		S	-	-	E		R		E
Vaccinated cases	G	I	A	N	-	N	H		M		-	S	L	E		-		-

**Table viruses-11-00108-t002b:** 

(b)										
		**NS1**						**NS1**		
**Mild**		K196	196E	**Total**		**Severe**		K196	196E	**Total**
**NA**	V263	64	13	77		**NA**	V263	32	9	41
	263I	2	8	10			263I	0	27	27
**Total**		66	21	87		**Total**		32	36	68

**Table 3 viruses-11-00108-t003:** Estimated mean quasispecies diversity comparison according to segment. Number of samples (n), mean diversity, and Mann Whitney Wilcoxon test *p*-value resulting from their comparison, when considering whether variants above 1% or 5% are reported in each case.

				Segment							
				1	2	3	4	5	6	7	8
Mild/Severe	Threshold	1	n	62/47	6/9	58/46	74/56	61/45	64/49	82/68	60/46
			Mean	1.88/2.38	2.33/2.55	2.03/3.46	0.99/1.54	2.75/2.73	1.64/2.24	0.41/0.65	0.42/0.59
			p	0.17	0.81	<0.001	<0.005	0.88	0.03	0.53	0.17
		5	n	80/69	53/38	80/67	85/69	81/65	83/68	91/73	83/68
			Mean	1.68/1.95	0.83/1.05	1.71/2.84	0.93/1.43	2.40/2.23	1.64/2.07	0.41/0.60	0.42/0.54
			p	0.40	0.26	<0.001	<0.005	0.73	0.27	0.75	0.25
Not Vaccinated/Vaccinated	Threshold	1	n	45/23	5/3	44/22	56/26	46/22	51/23	65/26	45/19
			Mean	2.22/1.78	3.4/1.3	2.84/2.13	1.20/0.77	3.22/2.27	1.84/1.47	0.51/0.35	0.44/0.47
			p	0.57	0.36	0.49	0.33	0.05	0.49	0.76	0.58
		5	n	62/26	37/17	62/26	67/27	63/26	65/27	69/27	65/25
			Mean	1.85/1.69	1.14/0.53	2.23/2.00	1.18/0.74	2.68/2.15	1.75/1.63	0.51/0.33	0.45/0.52
			p	0.76	0.26	0.89	0.28	0.30	0.77	0.62	0.47

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
