# Peer review of "Whole Genome Sequencing of A(H3N2) Influenza Viruses Reveals Variants Associated with Severity during the 2016–2017 Season"

_viruses, 2019, doi:10.3390/v11020108_

Round 1
Reviewer 1 Report
The manuscript by Simon et al., investigates the sequence diversity and clinical significance of mutations identified in clinical respiratory samples collected during the 2016-2017 influenza season in France. Specifically, RNA was extracted from clinical respiratory samples and all segments of the influenza virus genome were amplified by multi-segment PCR. Nucleic acids were sequenced using an Illumina platform NextSeq500 (2 x 150 nucleotides paired-end reads). Sequences were aligned and phylogenetic trees were created to determine significant sample mutations as well as relatedness within the sample set. SNPs and quasi-species diversity were analyzed using the authors’ developed algorithm. It was found that segment 2 (PB1) had lower sequence efficiency. All samples sequenced belonged to the 3C.2a subclades 3C.2a1, 3C.2a2, 3C.2a3, and 3C.2a4 with the exception of one sample which clustered with 3C.3a. The authors stratified the sample mutations by both disease severity and vaccination status to identify genomic viral changes that may be associated with increase pathogenesis or variant susceptibility. Several substitutions correlated with disease severity or vaccination status. The authors identified the substitutions V263I on NA and K196E of NS1 as being the most associated with severe cases. Analysis of quasi-species diversity showed that for the segments 3, 4, and 6, samples collected from severe cases had significantly increased viral population diversity within each sample. It is unclear if disease severity is a product of host factors allowing the increase of viral diversity or if viral diversity influenced disease severity. Taken together, the study shows interesting regional trends of influenza virus biology for the 2016-2017 influenza season but the paper could have included more graphical figures to help the reader grasp the mutations according to the protein structure. Furthermore, the manuscript needs significant editing for English.
Major Comments
1. The manuscript identified many mutations that could be stratified by patient factors (disease severity and vaccination status). It would be helpful if the authors included a 3D graphic and/or ribbon diagram (Pymol (http://www.pymol.org) or RCSB’s Protein Workshop (http://www.rcsb.org)) to identify placement of the mutations, especially in association with vaccination status and HA and NA as typically these would be the only proteins relevant due to vaccination preparations.
2. The manuscript needs significant editing for English specifically in the Introduction and Discussion sections. At times, I was unable to understand the point the authors were trying to get across. I have indicated specific sentences in the Minor Comments but the entire manuscript should be edited.
3. The introduction should include a summary of H3N2 circulation in humans in respect to 3C.2a and 3C.3a clades. Specifically, in the 3C.2a clade emerged in 2014-2015 season where A/Texas/50/2012 (clade 3C.1) was chosen as the vaccine reference strain. What is interesting and relevant to the present study is that the 3C.2a clade has a novel glycosylation site which is lost when the strain is grown in eggs. Propagation of the vaccine strain in eggs and loss of the glycosylation site changes the antigenicity of the vaccine compared to the circulating 3C.2a strains (Zost SJ et al., PNAS 2017). The subsequent vaccine strains for the following years and their respective clades were A/Switzerland/9715293/2013 clade 3C.3a (2015-2016); A/HongKong/4810/2014 clade 3C.2a (2016-2017 & 2017-2018); and A/Singapore/INFIMH/016-0019/2016 clade 3C.2a1 (2018-2019). All strains belonging to clade 3C.2a indicates the importance of 3C.2a to human influenza virus circulation and infection. Since 3C.2a strains represented the majority of the identified strains in the work, having a better background in the introduction would aid the translation of the work’s importance.
Minor Comments
Manuscript editing
1. The first sentence of the introduction is inaccurate (33-34). Influenza seasonality occurs not solely due to limited uptake of vaccines but also due to the incomplete effectiveness of vaccination and the antigenic variability of the pathogen. This sentence needs to be rewritten.
2. Lines 34-37 are confusing and need to be rewritten.
3. Line 42-43 is also inaccurate. The effectiveness of antivirals is contentious with many studies showing limited effectiveness and/or viruses have mutated to be resistant to antivirals (D. Cohen BMJ 2012). I don’t know of any antiviral that would be support the statement in 42-43.
4. Please reword line 49-49.
5. I don’t understand line 54-56.
6. Please change all mention of “vaccinal status” to vaccination status.
7. Please be consistent with the tables reporting numbers with decimal places. Either use all “,” or all “.” to represent the decimal.
8. I am confused by lines 272-274, 279-281, and 287-289.
Author Response
1. The manuscript identified many mutations that could be stratified by patient factors (disease severity and vaccination status). It would be helpful if the authors included a 3D graphic and/or ribbon diagram (Pymol (http://www.pymol.org) or RCSB’s Protein Workshop (http://www.rcsb.org)) to identify placement of the mutations, especially in association with vaccination status and HA and NA as typically these would be the only proteins relevant due to vaccination preparations.
à As suggested, we now include a figure showing the location of substitutions concerning HA, NA and NS1 on the closest H3N2 crystal structures we found, with some context of these proteins’ functional sites. We comment this point; in the results “Based on 3D structures, it could be noted that some of these substitutions were found: On HA positions 131 and 144 which are close to its receptor binding site (Figure 3A), on NS1 p85β binding site positions 99 and 146 (Figure 3B), and on NA position 149 which is close to its sialidase active site (Figure 3C).”; and in the discussion “We also reported substitution on V149A on NA which is close to the active site, may have some impact on the sialidase activity.”, “Notably, substitutions were found on position 131 and 145 close to positions 155 and 145 which have been shown to be crucial for the evolution of H3N2 viruses (Koel et al. Science 2013).”.
2. The manuscript needs significant editing for English specifically in the Introduction and Discussion sections. At times, I was unable to understand the point the authors were trying to get across. I have indicated specific sentences in the Minor Comments but the entire manuscript should be edited.
à We’re grateful for the help you provided by listing some of these issues in your comments. We had the manuscript checked by a native English speaker and changed several sentences following his instructions. These changes are tracked in the manuscript, so that they are easily visible for reviewing.
3. The introduction should include a summary of H3N2 circulation in humans in respect to 3C.2a and 3C.3a clades. Specifically, in the 3C.2a clade emerged in 2014-2015 season where A/Texas/50/2012 (clade 3C.1) was chosen as the vaccine reference strain. What is interesting and relevant to the present study is that the 3C.2a clade has a novel glycosylation site which is lost when the strain is grown in eggs. Propagation of the vaccine strain in eggs and loss of the glycosylation site changes the antigenicity of the vaccine compared to the circulating 3C.2a strains (Zost SJ et al., PNAS 2017). The subsequent vaccine strains for the following years and their respective clades were A/Switzerland/9715293/2013 clade 3C.3a (2015-2016); A/HongKong/4810/2014 clade 3C.2a (2016-2017 & 2017-2018); and A/Singapore/INFIMH/016-0019/2016 clade 3C.2a1 (2018-2019). All strains belonging to clade 3C.2a indicates the importance of 3C.2a to human influenza virus circulation and infection. Since 3C.2a strains represented the majority of the identified strains in the work, having a better background in the introduction would aid the translation of the work’s importance.
à We agree with your comment and have completed our introduction accordingly (line 67). This way, we hope to introduce these problematics sufficiently to highlight the importance of our work. “During the 2016-2017 influenza epidemic, the majority of influenza viruses circulating in the northern hemisphere was A(H3N2) viruse from clade 3c.2a. In France, the dominance of A(H3N2) viruses was associated with an early season of moderate intensity, but with a high severity particularly among the elderly in a context of low vaccine coverage and sub-optimal vaccine effectiveness (invs 2018). Low vaccine effectiveness was related to a hemagglutinin mutation in the 2016-2017 egg-adapted H3N2 (clade 3c.2a) vaccine strain A/HongKong/4801/2014 that altered its antigenicity (Zost SJ et al. PNAS 2017).”
Minor Comments
Manuscript editing
1. The first sentence of the introduction is inaccurate (33-34). Influenza seasonality occurs not solely due to limited uptake of vaccines but also due to the incomplete effectiveness of vaccination and the antigenic variability of the pathogen. This sentence needs to be rewritten.
à This sentence was changed to better reflect the root of this problem: “The Seasonal influenza infections remain a considerable health and socioeconomic challenge, as vaccines uptake and effectiveness are still limited.”
2. Lines 34-37 are confusing and need to be rewritten.
à These lines were rewritten to make them clearer: “The resulting morbidity and mortality have only marginally decreased in recent years, with annual epidemics still causing around 290,000 to 650,000 respiratory deaths worldwide [1]. This continuing burden leads to constantly adapt public health measures including vaccination policies, which remain the most potentially effective method to limit the impact of influenza [2].”
3. Line 42-43 is also inaccurate. The effectiveness of antivirals is contentious with many studies showing limited effectiveness and/or viruses have mutated to be resistant to antivirals (D. Cohen BMJ 2012). I don’t know of any antiviral that would be support the statement in 42-43.
à This sentence was changed to be more nuanced on the effectiveness of antivirals: “Antivirals that permit to control the infection in some patients at risk of complications are available and usually recommended, yet resistant strains often appear in cases of prolonged treatments.”.
4. Please reword line 49-49.
à This sentence was reworded as requested: “Many mutations in the influenza genome have been linked to virulence but clinical studies are scarce and have had, apparently, a minimal impact on clinical management, unlike what is routinely done in case of bacterial infections [8].”.
5. I don’t understand line 54-56.
à This sentence was rewritten for better understanding: “Therefore, global approaches considering virological and immunological parameters associated with clinical severity, as a whole, are better suited to investigate influenza pathogenesis [9,10].”
6. Please change all mention of “vaccinal status” to vaccination status.
à All mentions of “vaccinal status” were changed, as suggested.
7. Please be consistent with the tables reporting numbers with decimal places. Either use all “,” or all “.” to represent the decimal.
à The manuscript was corrected to use “.” as decimals consistently in the tables.
8. I am confused by lines 272-274, 279-281, and 287-289.
à These sentences were modified as follows:
Line 272-274 “It was also suggested that this site was part of an epitope that was suitable for therapeutic monoclonal antibodies, and that an amino-acid change at this position modified the immune response [35].”
Line 279-281 “NGS has brought to light new areas of research in microbiology, in particular on genetic diversity and on the implication of quasispecies in physiopathology [36–41].”
Line 287-289 “As explained in the sequencing efficacy description of the samples herein, an inconsistent read depth was observed on polymerase-coding segments, which could suggest the existence of defective particles in certain cases [15].”
Reviewer 2 Report
This study focuses on the difference of influenza genomic traits between mid and severe cases. 176 samples were collected and 155 H3N2 viruses were analyzed by deep sequence. Two important mutations were highly involved with severe outcomes. This study provides supportive information to help people understand influenza genetic traits.
1. In this study, the samples were collected from different ways including NPA, NS, TBA, and BAL. The distribution of sialic acid receptor is different between the upper respiratory tract and lower respiratory tract. This will affect the genetic mutation of the virus. 24 samples were collected from the lower respiratory tract in severe cases. This may cause an unfair conclusion. The author should discuss this.
2. Although the two substitutions (NA V263I and NS1 K196E) were highly enriched in severe cases, this may not affect the truth. Like , Hemagglutinin mutation D222N of the 2009 pandemic H1N1 influenza virus was highly enriched in severe outcomes in the clinic but did not affect virulence in mice (Virus research 189 (2014): 79-86). Generally, HA will play an important role to causes the lower vaccine efficacy. In this study, the author did not find some substitutions in HA which were highly involved with severe outcomes. Please explain this.
Author Response
1. In this study, the samples were collected from different ways including NPA, NS, TBA, and BAL. The distribution of sialic acid receptor is different between the upper respiratory tract and lower respiratory tract. This will affect the genetic mutation of the virus. 24 samples were collected from the lower respiratory tract in severe cases. This may cause an unfair conclusion. The author should discuss this.
à We completely agree with your comment; different sample nature may impact the mutations associated with diversity. However, including both kind of sample seemed necessary to remain comprehensive. In its current state, we mention in the manuscript line 247-248 the lack of multivariate statistical analysis taking into account sample nature as a limit of the study. However, we tried to minimize this bias by checking if substitutions or viral diversity were significantly associated to a group when comparing lower and higher respiratory tract samples of Lyon HCL cases, as mentioned lines 187-189, lines 230-232 and in the discussion lines 255-260. In addition, one might expect to find substitutions in the HA RBS if influenza viruses’ mutations were driven by location on the respiratory tract. Such mutations were not observed in this cohort. We added in the manuscript a sentence specifically on this problematic: “In addition, we did not find any substitutions in the HA receptor binding site contrary to what could be expected if influenza viruses’ mutations were driven by location on the respiratory tract”.
2. Although the two substitutions (NA V263I and NS1 K196E) were highly enriched in severe cases, this may not affect the truth. Like, Hemagglutinin mutation D222N of the 2009 pandemic H1N1 influenza virus was highly enriched in severe outcomes in the clinic but did not affect virulence in mice (Virus research 189 (2014): 79-86). Generally, HA will play an important role to causes the lower vaccine efficacy. In this study, the author did not find some substitutions in HA which were highly involved with severe outcomes. Please explain this.
à While most HA substitutions reported in our study were associated to vaccinated cases, some permitted to distinguish the severe cases from the mild ones. We report in table 2 that variants on positions 121, 131 and 142 were significantly less found among severe cases, albeit with a p-value barely significant. Interestingly, one minor variant (HA 160K) corresponding to the loss of a glycosylation site was reported significantly associated to severe cases in our study. We agree that the involvement of these substitutions on severity can’t be discussed extensively without further investigations similarly to what was done for H1N1 HA D222N substitution. Hence, we conclude in our paper that substitutions reported herein may be associated with severity but require further experimental validation.